# *Pseudomonas aeruginosa*: Infections, Animal Modeling, and Therapeutics

**DOI:** 10.3390/cells12010199

**Published:** 2023-01-03

**Authors:** Stephen J. Wood, Timothy M. Kuzel, Sasha H. Shafikhani

**Affiliations:** 1Department of Medicine, Division of Hematology, Oncology, & Cell Therapy, Rush University Medical Center, Chicago, IL 60612, USA; 2Department of Microbial Pathogens and Immunity, Rush University Medical Center, Chicago, IL 60612, USA; 3Cancer Center, Rush University Medical Center, Chicago, IL 60612, USA

**Keywords:** *Pseudomonas aeruginosa*, infection, acute infections, chronic infections, cystic fibrosis (CF), antibiotic resistance, virulence factors, animal modeling

## Abstract

*Pseudomonas aeruginosa* is an important Gram-negative opportunistic pathogen which causes many severe acute and chronic infections with high morbidity, and mortality rates as high as 40%. What makes *P. aeruginosa* a particularly challenging pathogen is its high intrinsic and acquired resistance to many of the available antibiotics. In this review, we review the important acute and chronic infections caused by this pathogen. We next discuss various animal models which have been developed to evaluate *P. aeruginosa* pathogenesis and assess therapeutics against this pathogen. Next, we review current treatments (antibiotics and vaccines) and provide an overview of their efficacies and their limitations. Finally, we highlight exciting literature on novel antibiotic-free strategies to control *P. aeruginosa* infections.

## 1. Introduction

Charles-Emmanuel Sédillot, a French military physician, was the first to reference an infection involving *Pseudomonas aeruginosa* in 1850. Sédillot described how surgical dressings of patients both in the field and hospital beds often became colored with a blue-green substance having a grape-like sweet odor, which we now know as the diagnostic hallmarks of *P. aeruginosa* infections, due to pyocyanin and 2-aminoacetophenone productions, respectively [1]. Thirty years later in 1882, a French pharmacist by the name of Carle Gessard reported the successful isolation of *P. aeruginosa* which he initially termed *Bacillis pyocyaneus*, in a publication entitled “On the blue and green coloration of bandages” [2,3]. It is not surprising that the site of the infection these authors described were wound dressings of surgical incisions given that wounds are preferred niches for *P. aeruginosa* infection [4,5,6,7,8,9,10,11,12,13,14]. After many taxonomic revisions over the past 100 years, at present, the *Pseudomonas aeruginosa* species is identified on the basis of 16S rRNA and genomic sequence comparisons, analysis of the cellular fatty acids, virulence factors, and differentiating physiological and biochemical tests [15,16,17,18,19,20,21,22,23,24,25,26]. 

*P. aeruginosa* is a rod-shaped Gram-negative bacterium of the class γ-proteobacteria and family Pseudomonadaceae [27]. It is a facultative aerobe that prefers to use oxygen as the final electron acceptor during aerobic respiration, although it is also capable of anaerobic respiration using other alternative electron acceptors such as nitrate [28]. *P. aeruginosa* can also catabolize a wide-range of organic molecules for nutrients, making it one of the most biochemically versatile and ubiquitous bacterium found in many environments such as soil, water, vegetation, and even human skin and oral mucosa [29,30,31]. The ability *P. aeruginosa* to thrive in diverse environments increases *P. aeruginosa* reservoirs and the possibility for exposure, leading to higher incidence of infections. *P. aeruginosa* has been isolated from locations such as hot tubs, humidifiers, and soil [1,32]. While in hospitals, *P. aeruginosa* has been isolated from respirators, physical therapy pools, sinks, and mops [33,34]. Patients infected with *P. aeruginosa* can also act as sources for new infections in hospitals [35]. These large reservoirs in both community and hospital settings allow for significant rates of infections. Because of its metabolic versatility and an arsenal of virulence factors it possesses [25], *P. aeruginosa* is responsible for many serious and life-threatening acute and chronic infections, particularly in the setting of immunocompromised hosts with mortality rates reaching as high as 40% [36,37,38,39]. 

*P. aeruginosa* is a killer of immunocompromised patients, a leading cause of bacteremia and sepsis in neutropenic cancer patients undergoing chemotherapy, and the number one cause of hospital-acquired pneumonia and respiratory failure [36,37,38,39]. *P. aeruginosa* infections are also common in diabetic ulcers, burn wounds, corneal ulcers, and surgical wounds [4,5,6,10,11,12,13,14,37]. Chronic infection by *P. aeruginosa* is a characteristic of individuals afflicted with cystic fibrosis (CF) and accounts for the pulmonary failure that leads to death in these individuals [40]. Life-threatening infections with *P. aeruginosa* are also becoming increasingly frequent in patients with AIDS [4,5,6,10,11,12,13,14,37,41,42,43]. Exacerbating the challenge with *P. aeruginosa* infections is this organism’s high intrinsic and acquired resistance to many current antibiotics [44,45]. Because of this challenge and the severity of infections caused by *P. aeruginosa*, it has been placed amongst the priority pathogens, known as the ESKAPE (*Enterococcus faecium, Staphylococcus aureus, Klebsiella pneumoniae, Acinetobacter baumannii, Pseudomonas aeruginosa*, and *Enterobacter* species), for which new antimicrobial development is urgently needed according to the World Health Organization (WHO) and the Center for Disease Control (CDC) [45,46,47]. In this review, we review the important acute and chronic infections caused by this pathogen. We provide an overview of current antibiotic treatments and their failure due to development of antibiotic resistance in *P. aeruginosa*. Finally, we discuss various animal models, developed to assess *P. aeruginosa* acute and chronic infections, and highlight exciting literature on the development of novel antibiotic-free strategies to combat *P. aeruginosa* infections.

## 2. *Pseudomonas aeruginosa* Infections

In this section, we discuss the various acute and chronic infections that are caused by *P. aeruginosa* (summarized in Table 1). 

### 2.1. Pseudomonas aeruginosa Acute Infections

*P. aeruginosa* can cause serious acute infections including, respiratory tract infections [48,49], hospital acquired pneumonia (HAP) and ventilator-associated pneumonia (VAP) [50,51], keratitis and corneal ulcers in contact lens wearing individuals [52,53], urinary tract infection (UTI), blood stream infections (BSIs) [54,55,56], osteomyelitis [57,58], and endocarditis [59,60]. Between 1997–2008, *P. aeruginosa* was reported to be responsible for 21.8% of the hospital acquired pneumonia (HAP) or the ventilator associated pneumonia (VAP), second only to *Staphylococcus aureus* which accounted for 28% [61]. A more recent report in 2016 also attributed similar HAP and VAP infection rates to *P. aeruginosa* [62]. In a meta-analysis of 11 studies of VAP cases after post-cardiac surgery, *P. aeruginosa* was the causative agent in 23.2%, followed by *S. aureus* (20.2%) [63]. A prospective study of adult patients with nosocomial pneumonia involving 75 hospitals in 11 countries between 2008–2009, found *P. aeruginosa* to be the leading cause of HAP (15.6%) and the second-leading cause of VAP (25.9%) behind only *Acinetobacter* spp. (35.6%) [64]. In a meta-analysis of 50 studies in China between 2010 to 2014, *P. aeruginosa* was responsible for 19.4% (95% confidence interval (CI) 17.6–21.2%) of all isolates in VAP and 17.8% (95% CI 14.6–21.6%) in HAP [65]. *P. aeruginosa* infections in VAP and HAP can be deadly with mortality rates ranging between 13% to nearly 50%. In one retrospective study involving 110 patients with *P. aeruginosa* VAP from 2008 to 2013 in Intensive Care Unit (ICU) in Italy, the mortality rate was reported to be 44.5%, the highest amongst all other pathogens [66]. The higher mortality rates have been associated with the extent of antibiotic resistance of infecting *P. aeruginosa* strains (such as multidrug resistance (MDR) clinical isolates) or specific virulence factors such as the type 3 secretion system (T3SS) in the clinical isolates [66,67,68,69]. Other co-morbidities, such as obesity, diabetes, and age have also been independently associated with higher mortality rates in VAP and HAP caused by *P. aeruginosa* [66,70,71]. 

*P. aeruginosa* is also a leading cause of keratitis and corneal ulcers. In a meta-analysis report, covering publications in PubMed and Google Scholars (until 2016), the prevalence of *P. aeruginosa* in bacterial keratitis was shown to range from 6.8 to 55% [72]. In a multicenter retrospective study in Queensland, Australia between 2005–2015, involving 2176 patients with positive culture, *P. aeruginosa* was found to be the most prevalent Gram-negative pathogen, accounting for 17.7% of all infections [73]. In another multicenter study in Tehran, Iran, involving 6282 corneal scrapings from keratitis patients tested for infection, 2479 (39.5%) were culture positive and *P. aeruginosa* was found to be the most common cause of infection in patients with keratitis, although the Gram-positive *Streptococcus pneumonia* was the most prevalent isolated bacterium in keratitis patients older than 50 years [74]. In another study for the assessment of risk factors and the severity of disease in infection keratitis involving 231 patients, contact lens wear (53; 22%), ocular surface disease (45; 18%), ocular trauma (41; 16%), and prior ocular surgery (28; 11%) were found to be the major risk factors for infectious keratitis [75]. They also found *P. aeruginosa* to be the most prevalent cause of infectious keratitis and its presence was associated with significantly more severe keratitis [75]. *P. aeruginosa* keratitis has also been associated with worse outcomes and significant morbidity (i.e., worse initial visual acuity and size and extent of stromal involvement) in non-contact lens wearers in the elderly [76].

*P. aeruginosa* is also the third most common cause of urinary tract infections (UTIs), accounting for 7–15% reported infections [77,78]. In a study involving UTI in children, *P. aeruginosa* infection was associated with significant UTI recurrence, more resistance to antibiotic therapy, and longer hospitalization [78]. *P. aeruginosa* is a serious pathogen in the complicated UTIs, particularly in people with catheters, leading to life-threatening pyelonephritis [79]. Catheter-associated UTIs (CAUTI) account for nearly a million additional extra hospital days per year in the USA [80].

Bloodstream infections (BSIs) are amongst the most serious infections, with mortality rates ranging from 18% to 61% [81,82]. Not surprisingly, *P. aeruginosa* is also a major cause of BSIs. Results from a 13-year (2002–2015) prospective cohort study at Duke University Medical Center indicated significant increased mortality rate associated with BSI caused by *P. aeruginosa* as compared to other bacterial pathogens, including *Staphylococcus aureus* [83]. In the same study, the unadjusted time-to-mortality among patients with *P. aeruginosa* blood infection was also found to be significantly shorter than the patients with *S. aureus* bloodstream infection. The long-term sequalae of BSIs include very serious and life-threatening complications, such as elevated risks for venous thromboembolism and myocardial infarction and stroke [84,85], and neurocognitive disorders [86,87]. The SENTRY Antimicrobial Surveillance Program recently released a 20-year investigative report on the microbiology of blood stream infections from more than 264,901 BSI isolates collected from >200 medical centers in 45 nations between 1997 and 2016 [54]. *P. aeruginosa* was found to be the 4th leading cause of BSIs behind *S. aureus*, *E. coli*, and *K. pneumoniae*, accounting for 5.3% of all infections. Importantly, *P. aeruginosa* strains had the second highest incidence of MDR rates (26.3%). Collectively, these data highlight the deleterious impact of *P. aeruginosa* acute infections on public health, as has also been acknowledged by the World Health Organization (WHO) and The Center for Diseases Control (CDC) [45,46,47].

### 2.2. Pseudomonas aeruginosa Chronic Infections

Perhaps the most notable chronic infection caused by *P. aeruginosa* is the lungs of individuals with Cystic Fibrosis (CF) genetic disorder. *P. aeruginosa* is the leading cause of mortality and a major contributor to loss of lung function in people with cystic fibrosis (CF) [88,89,90]. The pathology associated with *P. aeruginosa* chronic infection in CF lung is in part due to collateral damage caused by infiltrating leukocytes as they unsuccessfully attempt to clear *P. aeruginosa* infection from the lungs of these individuals [88,89]. Colonization with *P. aeruginosa* occurs early in the life of CF patients, either through community or hospital exposure, and remains chronic throughout their lifespan [91]. 

Wounds are also highly vulnerable to *P. aeruginosa* infections. *P. aeruginosa* is the most abundant and frequently reported Gram-negative pathogenic bacterium in all chronic wounds—(wounds that do not heal within 3 months)—including diabetic foot ulcers, venous leg ulcers, and pressure ulcers [13,92,93,94,95,96,97,98]. Moreover, presence of *P. aeruginosa* in these wounds correlates with poor prognosis for healing [11,12,13,14,99]. *P. aeruginosa*-infected wounds have been reported to be significantly larger than wounds containing other bacteria—including *Staphylococcus aureus*, which is the most abundant Gram-positive bacterial pathogen reported in these wounds [11,13,100,101,102]—suggesting that *P. aeruginosa* infection may be more detrimental to the process of tissue repair and wound healing than other pathogens. Corroborating these reports, *P. aeruginosa* has been demonstrated to prevent wound healing and exacerbate tissue damage in cell culture-based in vitro and in various in vivo animal wound infection models [99,103,104,105,106]. In fact, a number of in vitro and in vivo studies indicate that wound is a preferred niche for *P. aeruginosa* [4,5,6,10,106].

### 2.3. P. aeruginosa Infections in Immunocompromised Patients

*P. aeruginosa* is one of the most commonly isolated Gram-negative bacterial pathogens responsible for severe infections in immunocompromised patients, such as HIV/AIDS patients, neutropenic cancer patients undergoing chemotherapy, or immunosuppressed hematopoietic stem cell transplantation (HSCT) patients [107,108,109,110,111,112,113,114,115]. *P. aeruginosa* has been reported to be the causative pathogen in 8 to 25% of community acquired pneumonia, sepsis, and UTIs in HIV patients [43,81,116,117,118,119,120]. The incidence of *P. aeruginosa* related-bacteremia was reported to be 10 times greater in patients infected with HIV, due to their immunocompromised condition [121]. 

Similarly, *P. aeruginosa* causes a variety of important infections (e.g., pneumonia, blood stream infections, UTI, and wound) with high morbidity and mortality rates in patients suffering from drug-induced neutropenia (reduced neutrophil count), drug-induced qualitative neutropenia (defects in neutrophil function), or drug-induced immunosuppressed patients [115,122,123]. *P. aeruginosa* infections are reported to be more common in neutropenic patients with malignancy, particularly those with leukemias [124,125]. Over 21% of bacteremia in patients with acute leukemia were reported to be due to *P. aeruginosa* infection [126,127]. *P. aeruginosa* was also the most common cause of pneumonia in patients with solid tumors [125]. 

Solid organ transplant (SOT) recipients are another immunocompromised group that is highly vulnerable to *P. aeruginosa* infections (particularly UTI, pneumonia, and bacteremia) following transplant [128,129,130]. The primary reason for the vulnerability of this group to infection is the use of potent immunosuppressive drugs to prevent transplant rejection [131]. The morbidity rates associated with BSIs in SOT recipients have been reported to range from 4.8–11% in kidney, 8–24% in heart, 8–25.7% in lung, 61–69% in intestinal, and 10–34% in liver transplant recipients, respectively [132,133,134,135]. Moreover, because prophylactic antibiotics are routinely administered in this group, the infecting pathogens tend to be highly resistant to antibiotics [128]. A recent prospective study assessed the burden and timeline of infectious diseases in the first year after solid organ transplantation among SOT recipients [136]. Out of 2761 SOT recipients, 1520 patients (55%) suffered 3520 episodes of infections, of which, bacterial infections accounted for 63%. *P. aeruginosa* was responsible for 9% of all infections and 23% of *P. aeruginosa* clinical isolates were MDR. In another study involving 191 episodes of BSI in SOT recipients, *P. aeruginosa* accounted for 5.2% of BSIs [137]. Mortality rates amongst SOT recipients infected with *P. aeruginosa* was shown to be between 33–40%, highlighting the significance of *P. aeruginosa* infection in this cohort of patients [129,130]. 

Another immunocompromised patient group that is highly susceptible to *P. aeruginosa* infection are the burn patients. Hyperinflammatory cytokine response and hypo innate and adaptive immunity are the hallmark of immunosuppression following major burn traumas in human [138,139,140]. *P. aeruginosa* has been reported as the most frequently recovered infectious pathogen in burn units and the clinical isolates display high resistance to antibiotics. In a recent large study involving 17119 patients at a burn ward in China between 2006 and 2019, *S. aureus* and *P. aeruginosa* were the predominant clinical pathogens responsible for bacterial infections in these patients [141]. It is worth noting that while the rates of *S. aureus* infections appeared to remain stable during this study (17.06% in 2006 to 18.54% in 2019); the rates of *P. aeruginosa* infections rose from 10.59% in 2006 to 17.68% in 2019, suggesting an upward trajectory in *P. aeruginosa* infections among burn patients at least in this burn unit. In another retrospective study assessing bacteria from wounds, catheters, blood, feces, urine and sputum of 10,276 hospitalized patients in burn wards between 2007 and 2014 in China; 3005 pathogenic strains were isolated and identified [142]. While *S. aureus* was the predominant strain in the beginning, its annual detection rates declined significantly over these years. In contrast, the annual detection rates of *P. aeruginosa* increased significantly during this period. Alarmingly, the detection rate increases in *P. aeruginosa* were associated with increased incidence of MDR bacteria in the culture, prompting the authors to caution against the use of ciprofloxacin, ceftazidime and cefoperazone/sulbactam to counter the related increase in resistance levels in *P. aeruginosa*. In another study involving 184 positive cultures from burn patients in burn unit in Iran, 205 different bacterial strains were isolated and identified, of which *P. aeruginosa* was the most prevalent, accounting for 57% of all clinical isolates; followed by *Acinetobacter* (17%), *E. coli* (12%), *S. aureus* (8%) and other organisms (6%) [143]. Importantly, over 90% of *P. aeruginosa* isolates displayed resistance to gentamicin, ceftizoxime, carbenicillin, cephalothin and ceftazidime. *P. aeruginosa* has also been the primary infective agent reported in other studies involving burn patients [143,144,145,146,147]. 

Another group of immunocompromised patients who are highly vulnerable to *P. aeruginosa* infection, are the individuals suffering from primary immunodeficiency disorders (PIDDs). PIDDs is a group of 300 diseases caused by rare genetic disorders, such as Caspase Eight Deficiency State (CEDS), Autoimmune Lymphoproliferative Syndrome (ALPS), Chronic Granulomatous Disease (CGD), etc. [148,149,150,151,152,153]. *P. aeruginosa* has been reported to cause serious blood infections with high mortality rates in these individuals [149,150,151,152,153]. 

## 3. *P. aeruginosa* Infection Animal Models

In this section, we review the animal models which have been developed to study various infections caused by *P. aeruginosa* and assess the effectiveness of conventional and emerging therapies against this pathogen (summarized in Table 1). 

### 3.1. Acute and Chronic Pneumonia Infection Models

Animal models have been extremely useful in advancing our understanding of *P. aeruginosa* pathogenesis and for the development and the therapeutic assessment of new antibiotics or novel biologicals to control this pathogen. Although most studies involving *P. aeruginosa* infections rely on mouse or rat models due to the cost and availability of reagents, larger animal modeling is also performed usually to fulfill the requirement by organizations—such as Food and Drug Administration (FDA)—to evaluate the efficacy and the safety profiles of new investigative biologics in two animal models that approximate human responses with respect to the condition under investigation [154]. As was discussed above, *P. aeruginosa* is an important bacterial pathogen in acute and chronic pneumonia, including the ventilated-associated pneumonia (VAP) and hospital acquired pneumonia (HAP). The first animal model of chronic pulmonary infection was a rat model in which *P. aeruginosa* infection was initiated by intratracheal inoculation of *P. aeruginosa* bacteria enmeshed in agar beads [155]. In this chronic model of infection, *P. aeruginosa* was detected during the 35 days of observation. Importantly, infected lungs in these rats exhibited lesions resembling those seen in lung tissues of humans with acute or chronic *P. aeruginosa* pneumonia, including the presence of goblet-cell hyperplasia, focal areas of necrosis, and acute and chronic inflammatory infiltrate [155]. Since then, various similar animal models (in mouse, rat, rabbit, porcine, dog, cat, etc.) of *P. aeruginosa* infections for acute and chronic pneumonia (including VAP and HAP) have also been described, albeit with modifications in the *P. aeruginosa* strain, initial inoculum levels, and in the rout of *P. aeruginosa* delivery into animal [156,157,158,159,160,161]. 

### 3.2. Urinary Tract and Kidney Infection Models

*P. aeruginosa* also causes urinary tract and kidney infections as discussed above. The initial animal models to assess UTI caused by *P. aeruginosa* involved intravenous injection of *P. aeruginosa* into mice [162,163]. In these systemic infection models, large doses (close to lethal doses) of *P. aeruginosa*—were needed to establish infection in the urinary tract and kidney. However, the high rate of mortality, due to systemic infection, made these murine models impractical [162,163]. To overcome this difficulty, artificial manipulations, (e.g., administration of bromoethylamine hydrobromide or ferric sorbitol citrate), were used to make kidneys in animals more susceptible to *P. aeruginosa* colonization and growth [164,165]. However, these artificial means made the interpretation of the data unreliable [166]. These limitations then led to the development of methods (e.g., surgical implantation of glass beads laden with *P. aeruginosa*, or transvesical ureteral catheterization) to directly deliver *P. aeruginosa* into the rat kidney in order to cause infection in this organ [167,168,169]. At present, urinary tract and kidney infection models frequently instill bacteria into the bladder using a catheter, based on the UTI protocol that was developed for Uropathogenic *Escherichia coli* (UPEC) by Hung et al. [170,171]. 

### 3.3. Blood Stream and Systemic Infection Models

Different approaches have been used to cause blood and systemic infection in animals with *P. aeruginosa*. For example, intravenous (i.v.), intraperitoneal (i.p.), or tail vein injections have been used as technical means to cause systemic infection with *P. aeruginosa* [172,173,174]. *P. aeruginosa* has also been delivered retro-orbitally to cause systemic infection and sepsis [175]. *P. aeruginosa* systemic infection has been shown to increase pro-inflammatory cytokines both in the blood and tissues, leading to other morbidities such as septic arthritis and gallbladder damage [172,176,177]. 

### 3.4. Keratitis and Corneal Ulcers Infection Models

As this was discussed above, keratitis and corneal ulcer infections are relatively rare but they are very serious medical conditions requiring urgent medical care because of the possibility that they can lead to vision loss in the affected eye(s). In the murine models for corneal ulcer, 2–3 parallel scratches (~1 mm) are usually made by sterile 25-gauge needle on the cornea of anesthetized animal prior to bacterial inoculation [178,179,180,181]. Animal models have been informative in showing the potency of antimicrobial activities in human tear [178]; in establishing the crucial roles for IL-16 pro-inflammatory cytokine and cathelicidin antimicrobial peptide in corneal defenses against *P. aeruginosa* [179,180]; and in demonstrating the therapeutic potential of the broad host range bacteriophage KPP12 in *P. aeruginosa* clearance and corneal healing [181]. 

### 3.5. Endocarditis Infection Models

Animal models of *S. aureus* infective endocarditis (IE) [182,183], are commonly used to investigate the underlying pathogenesis, disease progression, potential diagnostic approaches, and therapeutic treatment for endocarditis caused by *P. aeruginosa* [184]; Rabbits [185,186]. These models are based on surgical valve trauma followed by intravenous injection of bacteria within 10–24 h following the surgical valve trauma. In a rabbit model of endocarditis, with sterile right ventricular cardiac vegetations, Archer et al. demonstrated 78% mortality within 3 weeks, following *P. aeruginosa* infection [185]. In a follow-up study, the same group demonstrated that 14-day treatment with high dose gentamycin (7.5 mg/kg) and carbenicillin (400 mg/kg) was significantly more effective than either therapy alone, resulting in 64% sterilization of cardiac vegetations in this rabbit model of *P. aeruginosa* endocarditis [186]. In a rat model of *P. aeruginosa* endocarditis, Oechslin et al. demonstrated that a combination of systemic vancomycin and phage therapy was highly effective against *P. aeruginosa* endocarditis [184]. 

### 3.6. Wound and Surgical Site Infection Models

Skin is a formidable barrier against invading pathogens, including *P. aeruginosa* [187]. As an opportunistic pathogen, *P. aeruginosa* cannot colonize or cause infection in the skin of normal animal unless this barrier is breached by injury [4,5,6,10]. Therefore, animal wound and surgical site models of infections for *P. aeruginosa* (and other pathogens) usually involve full or partial thickness excisional wounding, or addition of bacteria directly to implants or stents before or after their insertion into animals [7,9,188,189,190,191,192,193,194,195]. In the settings of injury, *P. aeruginosa* can efficiently colonize and cause infection [11,12,13,14,196]. In a recent study, *P. aeruginosa* was shown to thrive in wound environment, (in a mouse model of wound infection), by dampening the host innate immune responses in wound tissue via inhibition of the NLRC4 inflammasome mediated by its most conserved virulence factor, ExoT [7]. 

Chronic wounds are particularly vulnerable to *P. aeruginosa* infection [100,197,198]. In a recent study involving db/db type 2 diabetic mouse, it was shown that impairment in the formyl peptide chemokine receptors (FPR) in diabetic neutrophils results in a delay in neutrophil response, rendering diabetic wounds vulnerable to colonization and infection by *P. aeruginosa* [8]. Macrophage response has also been shown to be delayed in db/db diabetic wounds, due to dysregulation in IL-10 expression and signaling [199,200], further dampening innate immune responses and diabetic wound’s ability to prevent *P. aeruginosa* infection [9]. In other studies, *P. aeruginosa* has been demonstrated to several other virulence factors (e.g., biofilm, type 3 secretion system (T3SS), pyocyanin, extracellular proteases, and Exotoxin A) to prevent wound healing and exacerbate tissue damage [99,103,104,105,106,201,202]. In a burn wound model of infection, *P. aeruginosa* infection was shown to lead to bacteremia in a manner that was dependent on superoxide response regulator (*soxR*) expression and function in *P. aeruginosa* [203]. In another report, quorum sensing (QS) was shown to be involved in biofilm maturation and *P. aeruginosa* colonization and pathogenesis a pressure ulcer infection model in rat [204]. 

### 3.7. Immunocompromised Infection Models

As was discussed above, immunocompromised people are highly vulnerable to infection with *P. aeruginosa*. Not surprisingly, animal models have been developed to assess the impact of *P. aeruginosa* infection in immunocompromised hosts. For example, Takase et al. demonstrated that *P. aeruginosa* infection in the calf muscle of immunocompromised mice, (generated by cyclophosphamide), caused high mortality in these mice, in a manner that was mediated by pyoverdine and pyochelin siderophore production in *P. aeruginosa* [205]. In another study, *P. aeruginosa* was shown to induce death within 46 to 59 h in a leukopenic immunosuppressed mouse model [206]. Similarly, Mahmoud et al. demonstrated that wounds in the neutropenic immunocompromised C57BL/6 mice are vulnerable to *P. aeruginosa* enhanced infection [188]. In another study, *P. aeruginosa* infection was proved to be highly lethal in an immunosuppressed guinea pig model of pneumonia [207].

### 3.8. Cystic Fibrosis Infection Animal Models

Cystic Fibrosis (CF) is a genetic disorder caused by null mutations in the cystic fibrosis transmembrane conductance regulator (*CFTR*) gene, which encodes for the chloride channel [208,209,210]. Not surprisingly, different transgenic animal species, (i.e., mice, rats, rabbits, ferrets, pigs, and sheep), harboring similar mutations in the CFTR gene have been constructed to model various CF pathologies [211,212,213,214,215,216]. In one report, endobronchial infection with a mucoid *P. aeruginosa* strain was shown to elicit production of TNF-α, MIP-2, and KC/N51 inflammatory cytokines in bronchoalveolar lavage fluid and cause 80% mortality in CF mice (harboring the S489X mutation of the CFTR gene), thus phenocopying some of the CF hallmark pathologies observed in human [217]. Corroborating these studies, van Heeckeren et. al., demonstrated that infection with *P. aeruginosa* resulted in significantly higher mortality rates, weight loss, higher lung pathology scores, and higher inflammatory mediator and neutrophil levels in the lungs of CF mice as compared to wildtype littermates [218]. However, murine models do not completely develop human CF disease severity in the pancreas, lung, intestine, liver, and other organs [219,220], thus necessitating the need for the development of larger animal models for CF, such as newborn pigs and ferrets [214,215,216,219,221]. For example, CF pigs were demonstrated to develop airway inflammation, mucus accumulation, and impaired bacterial clearance [222]. CF pig lungs contained multiple bacterial species, suggesting impaired immune defenses against bacteria [222]. 

## 4. Current Treatments for *P. aeruginosa* Infections 

Here, we provide an overview of conventional antibiotic treatments for *P. aeruginosa* infections. Full description on these therapies can be found in the guidelines from the Infectious Diseases Society of America (IDSA) [223,224]. Regardless of the antibiotic agent(s) administered, patients infected with *P. aeruginosa* should be closely monitored as *P. aeruginosa* can rapidly acquire additional resistance mechanisms while exposed to antibiotic therapy as discussed in Section 4 below and discussed previously [44]. A summary of the antibiotic treatments, their limitations, and the mechanisms of resistance to these antibiotics can be found in Table 2.

### 4.1. β-Lactam Antibiotics (Alone and Combination Therapies) 

β-lactam antibiotics generally are bactericidal antibiotics that destroy bacterial pathogens by disrupting their peptidoglycan cell wall via covalent binding to essential penicillin-binding proteins (PBPs) in bacteria [225]. Non-carbapenem β-lactam antibiotics (e.g., ceftazidime, cefepime, piperacillin-tazobactam, aztreonam, etc.) are the preferred first line of therapy where *P. aeruginosa* clinical isolate tests susceptible to these antibiotics [223,224]. The rational for their use is their clinical effectiveness using empiric regimes with fixed doses, their high potency and efficacy against a wide therapeutic range, and the low cytotoxicity and side-effects associated with their use [226,227]. Ceftazidime may be considered a preferred first-line β-lactam therapy because its use has been associated with the lowest risk of resistance while on therapy [228]. Carbapenems are a class broad spectrum β-lactam antibiotics broad spectrum of activity that are recommended to treat infections caused by bacteria, including *P. aeruginosa*, resistant to traditional β-lactams or fluoroquinolones (discussed below) [224,229]. 

Cefiderocol is a novel injectable cephalosporin which was approved by FDA in 2019 for the treatment of complicated UTIs [230]. It is also a preferred option for the treatment of uncomplicated and difficult-to-treat (DTR) cystitis caused by *P. aeruginosa* infection [223,224,231,232]. Cefiderocol has been shown to be more potent than both ceftazidime-avibactam and meropenem against all resistance phenotypes of *Pseudomonas aeruginosa* because of its unique siderophore-like property (which enhances its entry into the bacterial periplasmic space [233]) and because of its high stability to a variety of β-lactamases, including AmpC and extended-spectrum β-lactamases (ESBLs) [234,235].

More novel β-lactam combination antibiotic regiments (e.g., ceftazidime/avibactam, ceftolozane/tazobactam, imipenem/cilastatin/relebactam, etc.) are also used but only as alternative options for MDR or extensively-drug resistant (XDR) *P. aeruginosa* strains [223,224]. By definition, the MDR strains display acquired resistance to at least one agent in three antimicrobial categories, whereas the XDR strains show susceptibility to antibiotics in 2 or fewer categories and display resistance to at least one agent in the rest of available antibiotic categories [236,237]. Ceftazidime, (a third-generation broad-spectrum cephalosporin antibiotic [238]) is also used in combination with avibactam (a synthetic non–β-lactam, β-lactamase inhibitor which inactivates β-lactamase targets via covalent acylation [239]), and has shown to be highly effective against MDR and XDR *P. aeruginosa* strains (>90% effectiveness) [240,241]. However, *P. aeruginosa* strains resistant to ceftazidime/avibactam have been reported and are on the rise, adding to the challenges associated with *P. aeruginosa* therapeutics [242,243]. Ceftolozane, (a semi-synthetic broad-spectrum fifth-generation cephalosporin β-Lactam antibiotic [244]), in combination with tazobactam, (an irreversible β-lactamase inhibitor) [245], has been shown to be effective against most MDR and XDR resistant *P. aeruginosa* strains [245,246]. Not surprisingly, resistance to this combination therapy in *P. aeruginosa* has also been reported [247,248]. Combination therapy, imipenem (a β-lactam antibiotic belonging to the carbapenem subgroup [249]), cilastatin (an inhibitor of imipenem-degrading human renal dehydropeptidase and bacterial metallo-β−lactamase [250,251]), and relebactam (an inhibitor of β-lactamases [252]); has been recently approved in the USA and Europe, and shown to be highly effective for the treatment of complicated urinary tract infections including pyelonephritis, complicated intra-abdominal infections, and hospital-acquired bacterial pneumonia caused by MDR and XDR Gram-negative bacterial pathogens including *P. aeruginosa* [253,254,255]. Antibiotic resistance in *P. aeruginosa* will be discussed in the next section.

### 4.2. Fluoroquinolones

Fluoroquinolones (e.g., ciprofloxacin and levofloxacin) are broad-spectrum antibiotics that destroy bacteria by inhibiting DNA gyrase and topoisomerase IV (essential enzymes for DNA synthesis in bacteria) [256]. Fluoroquinolones are also recommended for the first line of treatment in acute otitis externa or skin and soft tissue infections and complicated UTI caused by *P. aeruginosa* strains that are found to be susceptible to these agents [223,224,257,258,259,260]. Limitations with the use of fluoroquinolones include rapid development of resistance in bacteria [261] and the sensitivity of fluoroquinolones to acidic conditions, which adversely affect their cell uptake [262,263], although recently developed novel fluoroquinolones (i.e., delafloxacin) show improved cellular uptake and maintain their antibacterial activities in acidic conditions [263].

### 4.3. Eravacycline (Tetracyclin) 

Generally, tetracycline antibiotics are not very effective against *P. aeruginosa* infections and may even enhance its virulence by stimulating its T3SS virulence function [264]. However, eravacycline (a novel fully synthetic fluorocycline) is different in that it exhibits potent activity against a broad spectrum of clinically relevant Gram-positive and Gram-negative aerobic and anaerobic bacteria, including *P. aeruginosa*, and has been recently approved in several countries for the treatment of complicated intra-abdominal infections in adult patients [265,266,267]. It owes its potency to 2 modifications in the tetracyclic *D* ring at position C7 (fluorine atom addition) and C9 (pyrrolidinoacetamo group addition) [268], which allow it to be effective even in pathogens that express tetracycline-specific efflux and ribosomal protection mechanisms in clinical isolates [269]. Like all tetracyclines, it exerts its antimicrobial activities by inhibiting protein synthesis machinery in bacteria through its interaction with the 30S (and to a lesser extent 50S) ribosomal subunits [266]. 

### 4.4. Aminoglycosides 

Aminoglycosides (e.g., gentamycin, amikacin, tobramycin) are broad-spectrum bactericidal antibiotics that destroy bacterial pathogens by targeting the 30S subunit of bacterial ribosomes, thus inhibiting protein synthesis [270]. They are also routinely prescribed against *P. aeruginosa* infections. A meta-analysis of publications until 2018 found that a single intravenous dose of aminoglycosides was highly effective (94.5 ± 4.3%) for uncomplicated cystitis with the recurrence-free and cure rate of >73%, with minimal toxicity [271]. In a large phase III clinical trial, plazomicin (a next-generation semisynthetic aminoglycoside) was shown to be noninferior to meropenem in the treatment of complicated urinary tract infections, even against Gram-negative pathogens, (including *P. aeruginosa*) with aminoglycoside-modifying enzymes that impart resistance to most aminoglycosides [272]. Other reports have confirmed the effectiveness of plazomicin against hard-to-treat resistant Gram-negative pathogens [273]. In another randomized trial involving seven U.S. centers for the treatment of cystic fibrosis, the short-term aerosol administration of a high dose of tobramycin in patients with clinically stable cystic fibrosis was shown to significantly increase lung function (as assessed by forced expiratory volume, forced vital capacity, and forced expiratory flow at the midportion of the vital capacity), and decrease the density of *P. aeruginosa* in sputum by a factor of 100, without any ototoxicity nor nephrotoxicity [274]. Although, nebulized amikacin/fosfomycin was shown to be ineffective in improving clinical outcomes in pneumonia in 3 clinical trials, despite reducing bacterial burden in these patient cohorts [275,276,277]. 

### 4.5. Polymyxins (Colistin, Polymyxin B) 

Therapeutic use of polymyxins as one of the last lines of therapy against MDR/XDR *P. aeruginosa* infections have skyrocketed during the past decade [278,279]. Polymyxins kill Gram-negative bacterial pathogens by disrupting their membrane via targeting the Lipid A moiety of the lipopolysaccharide (LPS) in the outer membrane of these bacteria [280]. Therapeutic use of polymyxins has been associated with relatively higher toxicity side-effects, (e.g., nephrotoxicity and neurotoxicity), as compared to other antimicrobials [281,282], thus dosing and the treatment protocol should be carefully designed to reduce these side-effects [283]. IDSA recommends polymyxin B (over colistin) in combination with the β-lactam and β-lactamase inhibitor for the treatment of non-urinary tract infections, if no aminoglycoside is effective against *P. aeruginosa* isolate in vitro [223,224]. The rational for this recommendation is that polymyxin B is not administered as a prodrug and therefore can achieve more reliable plasma concentrations than colistin which is administered as a prodrug and it converts to its active form in the urinary tract [284]. For precisely the same reason, colistin (not polymyxin B) monotherapy has been recommended as an alternate consideration for treating urinary tract infections caused by MDR/XDR *P. aeruginosa* [279], although IDSA also warns that clinicians should remain cognizant of the associated risk of nephrotoxicity in these patients [223,224]. 

## 5. Antibiotic Resistance in *P. aeruginosa*

The rate of antimicrobial resistance among *P. aeruginosa* clinical isolates has climbed sharply over the past 5 decades worldwide [285,286,287,288]. *P. aeruginosa* clinical isolates frequently exhibit resistance to various classes of antibiotics including β-lactams, aminoglycosides, fluoroquinolones, and even polymyxins with strains isolated from the Intensive Care Units (ICUs) demonstrating the highest incidence of resistance to these antibiotics [289,290,291]. A retrospective 10-year study reported a 10% increase in antibiotic resistance rates in *P. aeruginosa* clinical isolates in ICU within a decade [292]. Another study investigated the profile of antimicrobial resistance of Gram-negative bacteria in blood cultures in a university-affiliated hospital in China and found resistance rate to carbapenem among blood culture isolates of *P. aeruginosa* to increase significantly from 2004–2011 [293]. Of note, between 2000–2010, consumption of antibiotics (particularly carbapenems) increased by nearly 35% in China [294], thus establishing a direct correlation between antibiotic use and the emergence of antibiotic resistance in *P. aeruginosa* clinical isolates. Additionally, the incidence and the rate of infections with the multidrug resistant (MDR) *P. aeruginosa* strains—(resistant to at least one antibiotic in 3 or more antibiotic classes)—have been steadily rising over the past 20 years [286,295,296,297,298], as a direct consequence of increased antibiotic consumption [294]. A meta-analysis report assessing the clinical and economic impacts of hospital-acquired resistance and MDR *P. aeruginosa* infections between the years 2000 to 2013, indicated a greater than 2-fold increased risk of mortality in patients infected with MDR *P. aeruginosa* strains and a 24% increased risk of mortality in patients infected with resistant *P. aeruginosa* strains as compared to patients infected with antibiotic susceptible *P. aeruginosa* strains [297]. Not surprisingly, the same study found longer hospital stay and increased cost associated with resistant and MDR *P. aeruginosa* infections as compared to susceptible *P. aeruginosa* and control patients. Corroborating these data, an international multicenter retrospective study in 2015 reported significant increased association between mortality in nosocomial pneumonia and MDR *P. aeruginosa* infections [299]. 

Another frightening concern is the emergence of extensively drug resistant (XDR) *P. aeruginosa* infections among HAIs. XDR *P. aeruginosa* strains are defined as the strains that remain susceptible to only one or two classes of anti-Pseudomonas drugs and show resistance to at least 1 antibiotic in the other antibiotic groups [300,301]. A retrospective study of a cohort of adult, hospitalized patients with *P. aeruginosa* in Thailand reported that 22% of *P. aeruginosa* infections were due to XDR strains, resulting in significantly higher mortality [302]. Another prospective study involving 1915 ICU patients between 2014–2015 in India reported 47.7% of *P. aeruginosa* infections to be due to MDR and XDR *P. aeruginosa* strains [303]. Recently, a nosocomial outbreak of *P. aeruginosa* infection with XDR strains was reported among ICU patients subjected to aromatherapy in Austria, which was successfully addressed through the implementation of restriction of oil sharing among patients [304]. A systemic review and meta-analysis of 54 articles between 2000 and 2016 reported prior use of antibiotics and prior hospital or ICU stay as the most significant risk factors for nosocomial infections with MDR and XDR *P. aeruginosa* strains [305].

The emergence of multi-drug resistant *P. aeruginosa* poses even more serious threat in immunocompromised patients and complicates treatment options for these patients [306,307]. A retrospective study involving 7386 clinical specimens collected from HIV patients, reported increasing antibiotic resistance rates to aztreonam, cefepime, levofloxacin, meropenem, piperacillin, piperacillin-tazobactam, ticarcillin, and tobramycin in *P. aeruginosa* clinical isolates in this patient cohort [308]. Similar to HIV patients, in a 10-year study involving 149 solid organ transplant recipients, 43% of *P. aeruginosa* clinical isolates from transplant recipients were reported to be MDR [309]. 

## 6. The Mechanisms of Antibiotic Resistance 

The mechanisms of antibiotic resistance in *P. aeruginosa* have been reviewed extensively elsewhere [310,311,312,313]. In brief, resistance to antibiotics in *P. aeruginosa* is multifactorial involving various intrinsic (inherent) and extrinsic (acquired) mechanisms. The intrinsic mechanisms of antibiotic resistance in *P. aeruginosa* include: (i) expression of porin molecules (e.g., OprF) that are considerably more restrictive to antibiotics entry into *P. aeruginosa* as compared to other Gram-negative bacteria such as *E. coli* [314,315]; (ii) reduction in the expression of outer-membrane porins, which further renders bacterial cell wall less permeable to antibiotics [313,316]; (iii) expression of various efflux pumps (e.g., MexAB-OprM, MexCD-OprJ, MexEF-OprN, and MexXY-OprM), which impart resistance to β-lactams, fluoroquinolones, and aminoglycosides by pumping out these antibiotics thus reducing their effective concentrations in *P. aeruginosa* cytosol [313,317,318,319]; (iv) biofilm production which imparts resistance to antibiotics through various mechanisms—(discussed in [320,321,322,323,324])—including (a) biofilm-specific protection against oxidative stress induced by bactericidal antibiotics, (b) biofilm-specific expression of efflux pumps; (c) the protection provided by matrix polysaccharides through chemical interactions between the chemical functional groups in biofilm and antibiotics, thus limiting their accessibility to biofilm bacteria, and (d) the reduced metabolic state of biofilm bacteria; (v) emergence of persister and antibiotic tolerant phenotypes—(due to metabolic slowdown, oxygen limitations, and stress conditions)—that substantially increases resistance to antibiotics in otherwise sensitive bacteria, particularly in chronic infections [325,326,327,328,329]; (vi) mutations and polymorphism in the gene targets of antibiotics which reduce drug/target interactions, such as, quinolone resistance due to mutation in DNA Gyrase gene [330,331,332]. 

The extrinsic mechanisms of antibiotic resistance include acquisition of resistance genes through horizontal gene transfer (HGT) mechanisms, such as transformation, transduction, conjugation, transposons, outer membrane vesicles (OMVs), and insertion elements [333,334]). These resistance genes can inactivate antibiotics by hydrolysis, such as metallo or extended spectrum β-lactamases; or by changing antibiotics through structural modifications, such as aminoglycoside modifying enzymes (e.g., acetyltransferases, nucleotidyltransferase, and phosphotransferases) [289,335]. 

## 7. Emerging Therapies to Combat *P. aeruginosa* Infections

As discussed above, antibiotics are routinely prescribed for the treatment of *P. aeruginosa* infections. While antibiotics have saved numerous lives over the past 9 decades, their use is not without its problems. First, the widespread use of antibiotics has led to an explosion of antibiotic resistance [336,337,338]. Second, currently, there is no antibiotic in the market that is effective against all bacterial pathogens, therefore, choosing the right antibiotic for prophylaxis use is crucial [339]. The choice of prophylaxis antibiotic is empirical in that it is based on the most probable cause of infection [340,341]. Prophylaxis antibiotics could fail if the patient encounters a different pathogen or a pathogen that is resistant to the administered antibiotic [342]. Third, antibiotics have many undesirable and dangerous side-effects; including nephrotoxicity, ototoxicity, hepatotoxicity, acute renal failure, and dysbiosis in the gut microbiota which itself has been associated with obesity, diabetes, and immunological and neurological diseases such as Parkinson disease [343,344,345]. Forth, prophylactic antibiotic use is associated with increased risk of infection with *Clostridium difficile* which is one of the deadliest causes of nosocomial infections, costing approximately $1.5 billion annually in USA alone [346,347,348,349]. Fifth, antibiotic use has been shown to interfere with healing processes at surgical sites [350]. In this section, we discuss the antibiotic-free strategies that are gaining tractions in dealing with *P. aeruginosa* infections and possible limitations associated with them (summarized in Table 3). 

### 7.1. Immune System-Based Approaches against P. aeruginosa Infections

Innate immune system possesses powerful cellular, (e.g., neutrophils and macrophages), and humoral (e.g., antimicrobial peptides and complement), components that destroy pathogens via many mechanisms; such as phagocytosis, bursts of reactive oxygen and nitrogen species (ROS and RNS), antimicrobial peptides (AMPs) and complement-mediated direct microbial killing via membrane attack complex (MAC), and indirect microbial killing through opsonization and phagocytosis, and neutrophil extracellular trap (NET) [351,352,353,354,355]. Below, we discuss the two main immune system-based approaches to control infection. 

#### 7.1.1. Antimicrobial Peptides (AMPs)-Based Approaches against Bacterial Infections

AMPs are natural peptide-based antibiotics which are expressed by almost all life forms, including humans [356]. Generally, AMPs destroy pathogens by attacking and permeabilizing their membranes but there are some AMPs that kill their targets via different mechanisms such as modulation of membrane fluidity, and inhibition of intracellular pathways such as DNA replication and protein synthesis [356,357]. AMPs have been studied as promising new therapies to combat infections and some of them are even in clinical trials [358,359]. In one cell-culture based study, LL-37 (also known as cathelicidin) or cecropin(1–7)-melittin A(2–9) amide (CAMA) AMPs were shown to reduce the minimum biofilm eradication concentrations (MBEC) against biofilm *P. aeruginosa* by 8-fold [360]. In another in vitro study, immobilized Melimine and Mel4 chimeric cationic AMPs were shown to reduce biofilm *P. aeruginosa* viability by 82% and 63%, respectively [361], highlighting their therapeutic potential against device associated *P. aeruginosa* infections, such as catheter-associated pneumonia. In a rat model of systemic infection with *P. aeruginosa*, the AMPs (Magainin II and Cecropin A) exerted strong antimicrobial activity and achieved a significant reduction in bacterial levels, plasma endotoxin, and TNF-α concentrations when compared with control and rifampicin-treated groups [362]. Rifampicin and Magainin II or Cecropin A combined therapies showed synergistic effect in reducing infection and mortality rates when compared with singly treated and control groups. In another set of in vitro and in vivo lung infections, peptide ZY4 (a cathelicidin mimetic) was shown to be highly effective against standard and clinical MDR *P. aeruginosa* and *Acinetobacter baumannii*, strains in a mouse septicemia infection model [363]. Importantly, ZY4 showed low propensity to induce resistance, eradicated biofilm bacteria, (even killing persister bacteria), and decreased susceptibility to lung infection by *P. aeruginosa* and suppressed dissemination of *P. aeruginosa* and *A. baumannii* to other organs in mice. However, notable limitations have also hampered AMPs’ therapeutic use (including AMPs’ inherent cellular and tissue toxicities, potential limitations with their spectrum of activities against pathogens, and emergence of resistance to AMPs [359,364,365].

#### 7.1.2. Immunomodulator-Based Approaches against *P. aeruginosa* Infections 

In this approach, immunomodulators—primarily proinflammatory cytokines, such as CCL3, or bacterial products, such as N-Formylmethionine-leucyl-phenylalanine (fMLF)—are used to mobilize and activate innate immune responses at the possible site(s) of infection [8,188,189]. Immunomodulator-based approaches to control infection theoretically have many advantageous over antibiotics, although they could also be used in combination with antibiotics to further enhance their effectiveness. First, it is highly unlikely for a pathogen to develop resistance to all antimicrobial weapons that our immune system has at its disposal, including phagocytosis, bursts of reactive oxygen species (ROS), hypochlorous acid (HOCl), neutrophil and macrophage extracellular traps and antimicrobial peptides as discussed above. Second, immunomodulator-based therapies would not be empirically based because, once mobilized to the site of infection, immune system could in theory be effective against majority of infections regardless of their origin (bacterial, fungal, or viral). Third, many pathogens, (including *P. aeruginosa*), possess stealth mechanisms that allow them to establish infection even in immunocompetent healthy individuals by dampening host’s immune responses [7,366,367,368,369,370]. Immunomodulators could potentially overcome at least some of these stealth strategies and fortify tissue’s defenses against invading pathogens by mobilizing immune responses to the site of these stealth pathogens. Forth, immunomodulators likely have fewer undesirable side-effects and may be safer than antibiotics. For example, it is unlikely that topical application of immunomodulators would result in development of bacterial resistance, gut dysbiosis, or organ damage (all side-effects associated with the use antibiotics as discussed above). 

In a wound model of infection in diabetic mice, Roy et al. demonstrated that one-time topical treatment with low level CCL3 reduced *P. aeruginosa* infection by >99% by enhancing the neutrophil response in diabetic wounds [8]. Importantly, CCL3 treatment did not lead to persistent non-resolving inflammation in diabetic wounds, which is the hallmark of diabetic foot ulcers [371,372]. Rather, inflammation subsided over time and CCL3-treated diabetic wounds healed substantially better even in the presence of *P. aeruginosa* infection, which has been shown to exacerbate healing and contribute to heightened inflammatory environment in diabetic wounds as they age and become chronic [9,103,373]. In another wound infection model in immunocompetent C57BL/6 mice, Mahmud et al. recently demonstrated that one-time topical applications of CCL3, fMLF, or LPS immunomodulators were as effective, (if not more), as prophylactic tobramycin, in reducing *P. aeruginosa* infection (by nearly 90%) in wound [188]. Interestingly, these immunomodulators did not adversely impact healing processes. Rather, they even modestly stimulated wound healing in this immunocompetent animal even in the absence of infection. In another study involving periprosthetic implant joint infection with *Staphylococcus aureus*, Hamilton et al. demonstrated treatment with fMLP significantly reduced *S. aureus* infection levels by >90%. In addition, fMLF therapy reduced infection-induced peri-implant periosteal reaction, focal cortical loss, and areas of inflammatory infiltrate in mice distal femora. Importantly, fMLP treatment reduced pain behavior and increased weight-bearing at the implant leg in infected mice. These reports highlight the therapeutic potential of immunomodulators in reducing infections and potentially as stimulants in wound healing even in uninfected animals. The possible drawback against this approach is their possible ineffectiveness in severely immunocompromised patients who may lack the ability to mount effective immune responses to immunomodulators. 

### 7.2. Phage-Based Therapeutics against P. aeruginosa Infections

Lytic bacteriophages (also known as phages) are viruses that selectively target and kill bacteria by lysis [374,375]. Phage effective killing of *P. aeruginosa* was first reported in 1957 by Kellenberger et al. [376]. Since then, various animal models and clinical studies have demonstrated the effectiveness of phages as alternative or adjunctive therapies with antibiotics to treat *P. aeruginosa* infections particularly when infections with MDR or XDR *P. aeruginosa* strains are suspected [374,377,378,379,380]. In a mouse burn wound model of fatal infection with *P. aeruginosa*, McVay et al. demonstrated that phage cocktail consisting of three different *P. aeruginosa* phages was effective in increasing survival rates by 87%, (when administered via intraperitoneal (i.p.) route), as compared to only 6% in the sham control group [379]. In a CF zebra fish model, phage therapy was shown to substantially lower bacterial burden, reduce proinflammatory response, and significantly decrease lethality, caused by *P. aeruginosa* infection [380]. In a case report involving a patient with a pan drug-resistant *P. aeruginosa* spinal abscess, local and intravenous injections of a purified personalized phage cocktail adjunct therapy with antibiotics was able to heal this patient, despite the strain’s resistance to all antibiotics including ceftazidime/avibactam, ceftolozane/tazobactam, and colistin [377]. In another case report involving prosthetic vascular graft *P. aeruginosa* infection, single application of phage OMKO1 and ceftazidime was sufficient to resolve infection with no signs of recurrence [381]. Despite these successful cases, clinical trials of phage therapies show low to moderate efficacy with large variations in infection clearance between subjects within the studies [382]. In addition, phage therapies also have their own set of limitations, including development of resistance to phages in bacteria, the requirement for high dose of viral particles, and the potential serious but not life-threatening side-effects [382,383,384]. 

### 7.3. Therapeutics Targeting P. aeruginosa Virulence Factors 

*P. aeruginosa* pathogenesis is due to an arsenal of cell-associated and secreted virulence factors [385]. Therefore, disarming *P. aeruginosa* by reducing its virulence in vivo is a logical and promising therapeutic strategy that is gaining momentum [386,387,388,389,390,391]. For example, the T3SS virulence structure is an essential virulence structure for T3SS-expressing *P. aeruginosa* strains during infection and without it, this organism is rendered avirulent and cannot cause disease [9,392,393]. In a recent study, INP0341, a T3SS inhibitor, was shown to attenuate corneal infection by *P. aeruginosa* in an experimental model of murine keratitis [394]. In another study, monoclonal antibodies against T3SS structural component PcrV provided strong prophylactic protection in several murine infection models and a post-infection therapeutic model [389]. Quorum sensing (QS) is another major virulence mechanism that regulates the expression of many virulence factors and biofilm in *P. aeruginosa* [395]. In one study, sitagliptin (a QS inhibitor) was shown to inhibit pyocyanin, hemolysin, protease, and elastase in addition to blocking swimming, swarming and twitching motilities, and biofilm formation in *P. aeruginosa* [390]. In another recent report, the FDA-approved drug allopurinol showed anti-QS activity against *P. aeruginosa* and reduced the infiltration of *P. aeruginosa* and leucocytes and diminished the congestion in the liver and kidney tissues of infected mice [396]. Other QS inhibitors have also shown effectiveness in the treatment of drug resistant *P. aeruginosa* [397]. In another report, inhibition of *Pseudomonas aeruginosa* secreted virulence factors reduced lung inflammation in CF mice [398]. Other virulence factors (e.g., pyoverdine, AlgR, CdpR, RpoN, CysB, and AnvM) have also been investigated as potential drug targets to inhibit *P. aeruginosa* virulence during infection [399].

### 7.4. Vaccine Development against P. aeruginosa

With the rapid rise in the MDR and XDR *P. aeruginosa* clinical strains, it is absolutely vital to develop effective vaccines against *P. aeruginosa*. To date, there have also been several vaccine candidates developed against *P. aeruginosa* membrane and LPS O antigens, outer-membrane porin proteins, the T3SS structural components (PcrV) and secreted effector peptides (ExoU), and combinations of these factors [400,401,402,403,404,405]. One study assessed the effectiveness of oral and intraperitoneal (i.p.) vaccinations with attenuated Salmonella enterica serovar Typhimurium SL3261 expressing *P. aeruginosa* serogroup O11 O antigen against *P. aeruginosa* infection in an acute fatal pneumonia model in BALB/c mice [406]. They reported that while both modes of immunization elicited O11-specific serum immunoglobulin G (IgG) antibodies, IgA was observed only after oral immunization. They further showed that oral vaccination significantly increased survival in an acute fatal *P. aeruginosa* pneumonia model. In another study, a multivalent live-attenuated mucosal vaccination (combining up to 4 attenuated strains having different LPS serogroups) elicited opsonophagocytic antibodies, which were directed not only to the LPS O antigens but also to the LPS core and surface proteins, correlated with protective immunity to *P. aeruginosa* lung infections [407]. In another study, vaccination with *Pseudomonas aeruginosa* outer membrane vesicles (PA_OMVs), formulated with aluminum phosphate adjuvant, was shown to reduce bacterial colonization, cytokine secretion and tissue damage in the lung tissue, thus protecting mice from lethal challenge of *P. aeruginosa* [408]. 

As for vaccine candidates against the T3SS, active and passive immunization with the PcrV antigen was shown to protect mice against *P. aeruginosa*-induced lung inflammation and injury and ensured the survival of challenged mice [403]. Interestingly, antibodies to PcrV also inhibited the translocation of type III toxins, thus reducing *P. aeruginosa* pathogenesis. In a follow up study, intramuscular injection of PcrV_NH_ (a chimeric derivative of PcrV which contains the N-terminal domain (Met1-Lys127) and H12 domain (Leu251-Ile294) of PcrV)—elicited a multifactorial immune response and conferred broad protection in an acute *P. aeruginosa* pneumonia model and was equally effective to full-length PcrV [405]. Moreover, passive immunization with anti-PcrV_NH_ antibodies also showed significant protection, at least based on inhibition of the T3SS and mediation of opsonophagocytic killing activities. An important limitation with T3SS-directed vaccines is the emergence of PA7-like clade of *P. aeruginosa* strains which lack T3SS [409,410,411]. 

Vaccine candidates against recombinant outer membranes OprF–OprI conjugates have been shown to be well-tolerated in healthy volunteers patients following mucosal administration [401]; or in patients with severe burns following intramuscular injection [402]. These vaccines have also been shown to elicit specific serum IgG and s-IgA antibodies in these patients, respectively [401,402]. Supporting these findings, in a clinical trial study involving 48 volunteers in six vaccination groups with either a systemic, a nasal, or four newly constructed oral live vaccines based on attenuated live Salmonella, expressing *P. aeruginosa* OprF-OprI recombinant as antigen, it was reported that while systemic and mucosal vaccines induced a comparable rise of serum antibody titers, a significant rise of IgA and IgG antibodies in the lower airways was only noted after nasal and oral vaccinations [412]. Disappointingly, in a randomized placebo-controlled phase II study in ventilated ICU patients, OprF–OprI vaccination (via intramuscular injection) produced a significant immunogenic effect (increases specific IgG), but it did not reduce infection rates [404]. Despite these considerable efforts, no vaccine has yet been found to be clinically efficacious against this pathogen. The main obstacles to achieve this goal include poor immunogenicity of the protective epitopes, a large variant subtype antigens, leading to high degree of serologic variability, a large genome facilitating adaptation to new environments, phenotypic plasticity between acute and chronic infection, and variations in animal responses to *P. aeruginosa* that make determination of the optimal vaccine formulations difficult from such studies [400,413]. 

### 7.5. Other Antibiotic-Free Therapies for P. aeruginosa Infection

#### 7.5.1. Silver

Silver has been recognized for its antimicrobial properties for centuries and because it is considerably less toxic to human cells than pathogens, it has been used in a variety of applications to prevent infection [414]. Silver has a broad spectrum of antibacterial, antifungal and antiviral properties. Many mechanisms have been postulated to underlie silver antimicrobial action, including cell wall and cytoplasmic membrane disruption, inhibition of protein synthesis machinery by denaturation of ribosomes, interference with ATP production and chemiosmosis, production of ROS, and interference with DNA replication machinery (reviewed in [415]). Silver has been shown to be very effective against both planktonic and biofilm *P. aeruginosa*, although higher silver concentrations are needed for it to be effective against biofilm *P. aeruginosa* [416]. In a murine model of wound infection, silver nanoparticles (AgNPs) were shown to reduce *P. aeruginosa* infection by nearly 5 log-orders as compared to control wounds on day 4 post-infection and stimulate complete wound closure by day 12, although AgNPs in combination with tetracycline was more effective [417]. In another study, silver nanocomposite was used to deliver two pharmaceutical compounds (alginate lyase and ceftazidime) to degrade the alginate and eradicate biofilm *P. aeruginosa* from the lungs. Silver nanocomposites displayed a high dispersity, good biocompatibility, and high ceftazidime-loading capacity, and a low pH-dependent drug release and degradation profiles [418]. Importantly, they were highly effective in eradicating *P. aeruginosa* from the mouse lungs and decreasing the lung injuries. In another wound infection studies in diabetic mice, combination of AgNPs—immobilized on chitin-nanofiber sheet (CNFS)—and weakly acidic hypochlorous acid (HClO) was shown to significantly reduce *P. aeruginosa* infection burden and stimulate healing in an otherwise impaired healing animal model [419]. Because of its potent antimicrobial property, silver is routinely used as an adjunct in wound dressings [420]. However, because it can also be toxic to keratinocytes, it also has the potential to impair healing [420]. The application of silver as an adjunct therapy on medical devices may not be suitable. In a large randomized clinical trial involving 1309 hospitalized patients, coating urinary catheter with silver was not only ineffective in reducing the incidence of bacteriuria, it actually increased infection rates in male patients, particularly with *S. aureus* [421]. 

#### 7.5.2. Honey

Honey has long been known for its antimicrobial and healing properties, dating back to ancient times, well before we gained the knowledge that microorganisms could cause infection and disease [422]. Honey exhibits a broad-spectrum of antibacterial activity against both Gram-positive bacteria and Gram-negative bacteria [423]. The antibacterial activity of honey has been attributed to several factors, including its high viscosity, its simple and complex sugar contents, its mild acidic nature, its hydrogen peroxide content, and its high levels of phenolic compounds with antimicrobial properties [424,425]. Lu et al. recently demonstrated that honey, at substantially lower concentrations compared to those found in honey-based wound dressings, inhibited *P. aeruginosa* biofilm formation and significantly reduced established biofilms [426]. In another recent investigation, Bouzo et al. demonstrated that no single component of honey can account for its total antimicrobial action against *P. aeruginosa* [427]. They further showed that honey affects the expression of many genes, particularly the genes involved in the electron transport chain maintenance, causing proton leakage across membranes, and inducing membrane depolarization and permeabilization in *P. aeruginosa*. A randomized clinical trial (105 patients) to compare a medical grade antibacterial honey (Medihoney™) with conventional treatments in wound care, reported clinical benefits from using honey in wound care with 23% fewer episodes of infection [428].

#### 7.5.3. Hyperbaric Oxygen Therapy (HBOT)

HBOT is used as an adjunctive therapy in the management of infections such as diabetic foot, osteomyelitis, gas gangrene, necrotizing fasciitis, and fungal infections [429]. The underlying mechanisms of HBOT’s bactericidal functions involve activation of innate immune system (via restoration of neutrophils’ bactericidal functions under hypoxic environment or diabetic condition), and enhancement of ROS production in targeted bacteria [191,429,430]. In another study to evaluate the therapeutic potential of HBOT as an adjuvant to tobramycin treatment, HBOT was able to significantly enhance the effect of tobramycin against aggregates of all the *P. aeruginosa* isolates from CF patients in vitro. The effect was attributed to increased O_2_ levels [431]. In experimental subcutaneous and pulmonary rat infection models, oxygen therapy was shown to significantly reduce *P. aeruginosa* titers in the blood and bronchial aspirates, and protected animals against infection-induced mortality and morbidity as compared to control group [432]. In combination with amikacin antibiotic, these effects were even more impressive, highlighting HBOT’s potential as an adjunctive therapy. Although the impact of HOBT on *P. aeruginosa* infection has not been directly examined, two systemic reviews of reports on randomized controlled trials comparing the effects of therapeutic regimens which include HBOT with those that exclude HBOT (with or without sham therapy)—have found HBOT to be beneficial for the treatment of necrotizing infections and to improve wound healing in diabetic ulcers and ischemic leg ulcers [433,434].

#### 7.5.4. Negative Pressure Wound Therapy (NPWT)

NPWT has been shown to reduce surgical site infections (SSIs) by reducing fluid accumulation within the avascular dead space in a closed wound [435,436]. NPWT was shown to reduce infection and decrease mortality in a murine model of burn wound sepsis with *P. aeruginosa* [437]. In a rabbit wound model of infection, NPWT was shown to be significantly more effective than the control treatment (a sterile gauze dressing) in reducing the expression of virulence factors and *P. aeruginosa* bacteria counts [438]. In another report, NPWT was shown to reduce the mobility of *P. aeruginosa* and enhance wound healing in a rabbit ear biofilm infection [439]. NPWT was also shown to inhibit the invasion and proliferation of *P. aeruginosa* in burn-wounded tissue and decrease mortality in a murine model of burn-wound sepsis [437].

## 8. Concluding Remarks

*Pseudomonas aeruginosa* is a serious pathogen that can cause deadly acute and chronic infections particularly in immunocompromised hosts and CF patients. What makes this pathogen so successful in colonizing diverse environments within its host are its large genome, which gives this pathogen the metabolic flexibility to quickly adapt to changes within its environment; and its large arsenal of cell-associated and secreted virulence factors, which protect this pathogen from recognition and attack by host’s immune responses. What is alarming about *P. aeruginosa* is its high intrinsic and acquired resistance to many available antibiotics which further highlights the need and the impetus for the development of novel strategies to deal with this organisms.

**Table 1 cells-12-00199-t001:** *Pseudomonas aeruginosa* infection types, prevalence, and animal models.

** Acute Infections **		
**Site**	**Reported Prevalence**	**Infection Model**
**Respiratory Tract Infections** [48,49]	See pneumonia & CF infections below	Various murine models lung infection [155,156,158,159,160,161,162,440]
**Hospital acquired pneumonia** [50]	21.8% [61,62], 15.6% [64] 17.8% [65]	Intratracheal inoculation in various animal models infection [155,156,158,159,160,161,162,440]
**Ventilator-associated pneumonia** [50]	23.2% [63], 25.9% [64], 19.4% [65]	Intratracheal inoculation in various animal models infection [155,156,158,159,160,161,162,440]
**Keratitis and corneal ulcers** [52,53]	6.8% to 55% [72,73,74,75,76]	Eye infection in murine models [178,179,180,181].
**Urinary tract infections** [54]	7% to 17% [77,78]	-Mouse intravenous injection [162,163] -Surgical implantation of bacteria coated beads or bladder catheterization [167,168,169,170,171]
**Blood stream infections** [54,55,56]	18% to 61% [81,82]	Bacteria injection via intravenous, intraperitoneal, or retro-orbital routes [172,173,174,175]
**Osteomyelitis** [57,58]	- 10.8% of all osteomyelitis [441] - 66% of all *P. aeruginosa* osteomyelitis were acute, 44% chronic osteomyelitis [442]	Chronic osteomyelitis animal murine model [443]
**Endocarditis** [59,60]	0.015% [444]; 3% [60]	Rats [184]; Rabbits [185,186]
** Chronic Infections **		
**Site**	**Reported Prevalence**	**Infection Model**
**Cystic Fibrosis** [88,89,90]	60% to 70% infections in adult CF [445,446,447]	Transgenic mutant CFTR mice, rats, rabbits, ferrets, pigs, and sheep animal models [156,211,212,213,214,215,216,217,219,220,221,222]
**Wounds** [13,92,93,94,95,96,97,98,448]	- Diabetic ulcers; 10% [449]; 14.3% [450];18.8% [451]; 29.8% [97] - Burn wounds; 12.4–57% [141,143,452,453,454]	Full thickness excision skin wounds and burn wounds in mice, rats, [8,9,104,188,199,200]
**Infection in Immunocompromised Patients** [102,103,104,105,106,107,108,109,110]	- 8% to 25% in HIV patients [43,81,116,117,118,119,455] - >21% in acute leukemia [126,127] - 9% of solid organ transplant infections [137] - 57% of major burn wounds [143]	Drug-induced and transgenic immunosuppression in rodents & guinea pig [205,206,207]

**Table 2 cells-12-00199-t002:** Conventional antibiotic treatments for *Pseudomonas aeruginosa* infections, their limitations, and mechanisms of resistance in *P. aeruginosa*.

Antibiotic Therapy	Target	Limitations & Resistance Mechanisms
**b-lactam antibiotics:** - Non-carbapenem b-lactam antibiotics [223,224,225,227,228] - Carbapenems [223,224] - Cephalosporins [223,224]	Peptidoglycan cell wall production via covalent binding of penicillin-binding proteins [225,456]	- Expression of antibiotic restrictive porins [314,315] - Reduced expression of outer-membrane porins reducing antibiotic permeability [313,316] -Expression of efflux pumps which reduce antibiotic concentration [313,317,318,319] - Biofilm protections against antibiotics [320,321,322,324] - Emergence of antibiotic tolerant persister bacteria [325,326,327,328,329] - Mutation of antibiotic targets [330,331,332] - Acquisition of resistance genes via HGT [289,333,334,335]
**Fluoroquinolones** [223,224]	DNA synthesis via inhibition of DNA gyrase and topoisomerase IV [256,457]	
**Tetracycline** [223,224]	Protein synthesis via inhibition of 30S and 50S ribosomal subunits [458,459]	
**Aminoglycosides** [223,224]	Protein synthesis via inhibition of 30S ribosomal subunit [460,461]	
**Polymyxins** [223,224]	Lipid A moiety in outer membrane LPS [462,463]	

**Table 3 cells-12-00199-t003:** Emerging non-antibiotic therapeutics for *Pseudomonas aeruginosa* infections.

Therapy	Target	Therapy imitations & Resistance Mechanisms
**Antimicrobial peptides (AMPs)**	Membrane integrity, DNA replication, protein synthesis [356,357]	Cellular toxicity, limited spectrum of activity, Multiple resistance mechanisms, including alteration in cell wall & degradation by proteases [359,364,365]
**Immunomodulators**	Activation of host cellular immunity [8,188]	No cellular toxicity; Resistance not reported but highly unlikely as they activate multiple immune responses [8,188,189]
**Phage-based therapeutics**	Membrane lysis [374,375]	- Low clinical efficacy, Development of resistance, & side-effects in patients [382,383,384]
**Therapies against Virulence factors**	- T3SS inhibition by small molecule or antibody [389,394] - Quorum sensing activity [390,394] - Secreted virulence factors [398,399]	Not reported but bacteria can potentially become resistant to these therapies in similar mechanisms to antibiotics
**Vaccines**	- LPS O-antigens [407] - Outer membrane vesicles [408] - PcrV (T3SS) [403,405] - OprF-OprI [401,402,404,412]	Vaccines have not been clinically effective, Variant subtype antigens and serologic variability, & animal model variability in determining formulations [400,404,409,410,411,413]
**Silver**	Various [415]	- Cytotoxic to keratinocytes [420] - Potentially ineffective as medical device coating [421]
**Honey**	Various [424,425]	Not Reported
**Hyperbaric oxygen therapy**	Activation of innate immunity and enhanced ROS production in bacteria [191,429]	Not Reported
**Negative pressure wound therapy**	Bacterial proliferation [435,436,437]	Not Reported

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
