# Peer review of "Pseudomonas aeruginosa: Infections, Animal Modeling, and Therapeutics"

_cells, 2023, doi:10.3390/cells12010199_

Round 1

Reviewer 1 Report

This is a comprehensive review on Pseudomonas aeruginosa infections, their risks, current therapy and emerging therapeutic approaches. The review also covers the various animal models of P. aeruginosa infections that are available for studies on the mechanisms underlying pathogenesis and evaluation of potential new therapies. The review is well structured, updated, and clearly presented. It includes a good amount of information and may be beneficial as a source of information for those interested in this topic. Some comments and suggestions that may improve the manuscript further are outlined below.

Specific comments

1.      Title: Too detailed, too long. Must condense. My suggestion: "Pseudomonas aeruginosa infections"

2.      Section 2, lines 62-68: This text is unrelated to the title of the section. I suggest to incorporate the text into section 1.

3.      Sections 2.1 and 2.3 comprise a lot of statistical information, which is difficult to comprehend. For clarity, I recommend summarizing the information in a Table. I also recommend adding a Figure, presenting in a graphic manner the common infections caused by P. aeruginosa and their consequences.  As presented, this part of the text is colorless, somewhat boring.

4.      Section 3.2: 'Urinary tract infection models': Lines 256-260 (References 157, 158) refer to "impractical" infection models (line 260). If this is the case, I recommend omitting this short paragraph completely.

5.      Subsection 3.3 is missing (3.4 follows 3.2). Please correct numbering.

6.      Sections 3.5 and 3.6 ("Wound and surgical sites infection models" and "Immunocompromised Infection models", respectively): the text in these subsections does not describe specific models, as the titles imply. This text should be moved to section 2. Alternatively, if some of the statements here present results derived from animal infection models, this should be stated directly, referring to the model and describing it as was done in preceding subsections.

7.      References: The authors should be far more selective in choosing the references. The current list of 427 citations should be cut by half (down to ~ 200 citations).

Minor comments:

1.      Line 17: replace "failure" by "potential", effectiveness/efficacy, etc.

2.      Line 91: 'in meta-analyses' or 'in a meta-analysis

3.      Line 267: replace "today" by currently, at present, presently (choose one).

4.      Line 287: add extracellular proteases.

5.      Line 321: replace 'illicit' by 'elicit'

6.      Line 403: replace 'it' by 'in'

7.      Line 483: replace 'retrospectively' by 'retrospective'

8.      Line 504: replace 'has been' by 'have been'

9.      Line 507: replace 'includes' by 'include'

10.  Line 510: replace 'render' by 'renders'

11.  Line 581: what is A. in 'A. humannii'. Please spell out.

12.  Line 671:    …'was shown to inhibit inhibited' - choose one.

13.  Line 807: 'decreases' should be 'decrease'

Author Response

We thank the reviewer for the thorough review of our manuscript and for his/her insightful comments, queries, & suggestions. This review was one of the best reviews we have ever had. Addressing these comments has significantly improved our manuscript. Below, we will provide our responses to reviewers' queries in a point-by-point format.

  1. Title: Too detailed, too long. Must condense. My suggestion: "Pseudomonas aeruginosa infections"

Response: We have reduced the title to “Pseudomonas aeruginosa: Infections, Animal Modeling, and Therapeutics” We believe that including “Animal Modeling, and Therapeutics” in the title makes our article more appealing to the readers who are seeking information in these specific areas.

  1. Section 2, lines 62-68: This text is unrelated to the title of the section. I suggest to incorporate the text into section 1.

Response: We appreciate the reviewer’s comment. We have moved this section into the Introduction section.

  1. Sections 2.1 and 2.3 comprise a lot of statistical information, which is difficult to comprehend. For clarity, I recommend summarizing the information in a Table. I also recommend adding a Figure, presenting in a graphic manner the common infections caused by P. aeruginosa and their consequences.  As presented, this part of the text is colorless, somewhat boring.

Response: We have done as suggested. We have added 3 Tables to the revised manuscript to summarize infections & animal models (Table 1), conventional therapies (Table 2), & emerging therapies (Table 3).

  1. Section 3.2: 'Urinary tract infection models': Lines 256-260 (References 157, 158) refer to "impractical" infection models (line 260). If this is the case, I recommend omitting this short paragraph completely.

Response: This information was provided specifically to highlight the early attempts at developing infection animal models of the urinary tract and kidney infections. We also highlighted why they failed and subsequent animal models. We believe that providing such information, (if available), provides the readers with an appreciation and perspective on the historical difficulties of animal modeling development in disease.   

  1. Subsection 3.3 is missing (3.4 follows 3.2). Please correct numbering.

Response: We apologize for this oversight. We have corrected the section and subsection numbering in the revised manuscript.

  1. Sections 3.5 and 3.6 ("Wound and surgical sites infection models" and "Immunocompromised Infection models", respectively): the text in these subsections does not describe specific models, as the titles imply. This text should be moved to section 2. Alternatively, if some of the statements here present results derived from animal infection models, this should be stated directly, referring to the model and describing it as was done in preceding subsections.

Response: The statements in this section were reflective of various animal wound and surgical site infection models. We have revised this section as suggested by the reviewer.

  1. References: The authors should be far more selective in choosing the references. The current list of 427 citations should be cut by half (down to ~ 200 citations).

Response: We appreciate the reviewer’s point but this is a comprehensive review article. These citations were included because they were discussed in this article.

Minor comments:

  1. Line 17: replace "failure" by "potential", effectiveness/efficacy, etc.

Response: We revised this word.

  1. Line 91: 'in meta-analyses' or 'in a meta-analysis

Response: Done

  1. Line 267: replace "today" by currently, at present, presently (choose one).

Response: replaced by “at present”

  1. Line 287: add extracellular proteases.

Response: Revised according to the reviewer's suggestion.

  1. Line 321: replace 'illicit' by 'elicit'

Response: Revised according to the reviewer's suggestion.

  1. Line 403: replace 'it' by 'in'

Response: Revised according to the reviewer's suggestion.

  1. Line 483: replace 'retrospectively' by 'retrospective'

Response: Revised according to the reviewer's suggestion.

  1. Line 504: replace 'has been' by 'have been'

Response: Revised according to the reviewer's suggestion.

  1. Line 507: replace 'includes' by 'include'

Response: Revised according to the reviewer's suggestion.

  1. Line 510: replace 'render' by 'renders'

Response: Revised according to the reviewer's suggestion.

  1. Line 581: what is A. in 'A. humannii'. Please spell out.

Response: Acinetobacter baumannii. Revised according to the reviewer's suggestion.

  1. Line 671:    …'was shown to inhibit inhibited' - choose one.

Response: Revised according to the reviewer's suggestion.

  1. Line 807: 'decreases' should be 'decrease'

Response: Revised according to the reviewer's suggestion.

Reviewer 2 Report

Wood and Shafikhani have written a broad review on P. aeruginosa pathogenesis, and current and novel therapeutics used to treat it. The strengths of this review are the detailed section on prevalence of P. aeruginosa in different infection settings and bodily sites, extensive discussion on current therapeutics and their limitations, and new strategies for managing infections that are currently in development. The document is well written and will serve as a useful resource. 

My suggestions are as follows: 

The section on taxonomy is rather brief despite being highlighted by the review’s title. This would be an opportunity for expansion, such as describing the differences between P. aeruginosa strains, one of which is mentioned later (PA7 clade, line 713). 

An interesting point the authors make is regarding how P. aeruginosa may be overtaking S. aureus as a leading causative agent in burn wounds (line 211 and following). This is an interesting and concerning trend that may be of interest to the field. I am wondering if the authors identified other studies that report similar trends. 

The animal model section could be expanded to include models for more of the infection types mentioned, such as corneal ulcers (mentioned later, in line 667) and endocarditis. 

The antibiotic resistance mechanism section (#6) is relatively brief and refers the reader to more in-depth reviews. Perhaps this section can be combined with the prior section (#5) on the problem of antibiotic resistance, in general. 

Author Response

We thank the reviewer for the thorough review of our manuscript and for his/her insightful comments, queries, & suggestions. Addressing these comments has significantly improved our manuscript. Below, we will provide our responses to the reviewer's queries in a point-by-point format.

1) The section on taxonomy is rather brief despite being highlighted by the review’s title. This would be an opportunity for expansion, such as describing the differences between P. aeruginosa strains, one of which is mentioned later (PA7 clade, line 713). 

Response: We appreciate the reviewer’s comment in this regard. Since there are recent review articles on this topic, we felt that it was more appropriate for us to cite these articles, as opposed to repeating them. Therefore, we removed the section on taxonomy from the text but included this brief discussion on the taxonomy in the introduction section. We also slightly expanded this portion to include the most recent and relevant citations for the readers to follow should they need more information on the taxonomy and the 3 described clades of P. aeruginosa strains.

2) An interesting point the authors make is regarding how P. aeruginosa may be overtaking S. aureus as a leading causative agent in burn wounds (line 211 and following). This is an interesting and concerning trend that may be of interest to the field. I am wondering if the authors identified other studies that report similar trends. 

Response: We found another recent large study involving samples from 17119 patients at a burn ward in China between 2006 and 2019. S. aureus and P. aeruginosa were the predominant clinical pathogens responsible for bacterial infections in these patients [136]. Importantly, while the rates of S. aureus infections appeared to remain stable (17.06% in 2006 to 18.54% in 2019), the rates of P. aeruginosa infections rose from 10.59% in 2006 to 17.68% in 2019, suggesting an upward trend in P. aeruginosa infections among burn patients at least in this burn unit. We have added this study to the revised manuscript.

3) The animal model section could be expanded to include models for more of the infection types mentioned, such as corneal ulcers (mentioned later, in line 667) and endocarditis. 

Response: We have added sections 2.4 and 2.5 in the revised manuscript to describe animal models in keratitis and corneal ulcers and endocarditis respectively.

4) The antibiotic resistance mechanism section (#6) is relatively brief and refers the reader to more in-depth reviews. Perhaps this section can be combined with the prior section (#5) on the problem of antibiotic resistance, in general. 

Response: We appreciate the reviewer’s comment in this regard but we believe that although this section is condensed, it does capture known mechanisms of antibiotic resistance in P. aeruginosa. In addition, we have included citations of recent review articles, should readers feel the need to seek more detailed information on these mechanisms of antibiotic resistance in P. aeruginosa.